# OpenReview forum: "ObjexMT: Objective Extraction and Metacognitive Calibration for LLM-as-a-Judge under Multi-Turn Jailbreaks"
_ICLR.cc/2026/Conference — Submitted to ICLR 2026_

### Official Review · Reviewer_Lrie · 2025-10-29

**Soundness:** 1
**Presentation:** 1
**Contribution:** 1
**Rating:** 0
**Confidence:** 5

**Summary:**

This paper proposes a benchmark for extracting latent objectives from multi-turn jailbreak conversations and evaluating model confidence calibration.

**Strengths:**

* Extracting true objectives from multi-turn adversarial conversations is genuinely challenging and practically relevant for AI safety.

**Weaknesses:**

* Related work is poorly organized and incomplete: Section 2 reads like a checklist of buzzwords rather than synthesizing relevant literature.
* Confidence motivation and calibration methodology are poorly justified and naive: The paper does not adequately explain why self-reported verbalized confidence is the right metacognitive signal—why not token probabilities or entropy-based uncertainty? The brief justification that verbalized confidence "sometimes outperforms token probabilities" doesn't explain when or why. The calibration methodology is naive: standard classification metrics (ECE, Brier) designed for probabilistic predictions are applied to self-reported confidence scores without validating that these have comparable semantics across models or that models interpret the elicitation prompt consistently. The paper treats verbalized confidence as ground truth about model uncertainty without empirical validation of this assumption.
* Unclear contribution: dataset repurposing or evaluation protocol? The paper claims to "introduce ObjexMT" but the three datasets already exist with ground truth base objectives—this is not a new dataset. The actual contributions are: (1) task formalization requiring single-sentence extraction + confidence, (2) LLM-judge evaluation protocol with threshold calibrated on only 300/2,817 items (10.6%), and (3) baseline predictions from 6 models.  With only 300 human labels and no inter-annotator agreement reported, this is insufficient validation for a benchmark. The paper needs to clearly articulate what is actually new
* The entire evaluation relies on GPT-4.1 as the sole judge to compute similarity scores, yet GPT-4.1 is itself one of the six models being evaluated. GPT-4.1 may be systematically favored, and results may not reflect true objective extraction quality but rather alignment with GPT-4.1's semantic representations.
* the paper is very unfinished.

**Questions:**

see weakness above

---

> ### Author Response · Authors · 2025-11-21
>
> 1. Section 2 is structured along five clear strands—LLM-as-a-judge, robustness under long and irrelevant context, multi-turn safety datasets, harmfulness discrimination vs. latent objective inference, and metacognition/calibration—so the characterization as a “checklist of buzzwords” does not accurately reflect its organization. That said, we will add a brief synthesizing paragraph in the revised version that ties these strands together and more explicitly summarizes ObjexMT’s position and differences relative to prior work.
>
> 2. Our goal is not to treat self-reported confidence as the ground truth of model uncertainty, but rather to measure how well this practically available signal aligns with correctness in realistic LLM-as-a-judge settings. Access to token-level log probabilities is uneven and, where present, often exposed through provider-specific APIs that are not directly comparable across all six frontier black-box models we evaluate. Some providers (e.g., OpenAI, Google’s Vertex Gemini, and DeepSeek) expose logprobs on at least some endpoints, whereas others (most notably Anthropic’s Claude APIs and some Kimi deployments) either do not expose token-level logprobs at all, or only make them available via non-standard proxy layers. Relying on logprob- or entropy-based uncertainty would therefore either (i) force us to exclude at least one major model from the benchmark, or (ii) require model- and provider-specific surrogate approximations, both of which are beyond the scope of this work. For this reason, we focus on a channel that can be elicited consistently from all models—verbalized self-reported confidence—and we cite prior work showing that such verbal confidence can in some cases be better calibrated than raw token probabilities.
> For calibration, we follow standard practice by treating each model’s scalar confidence (p \in [0,1]) as that model’s own probabilistic prediction, and we compute ECE, Brier score, and selective risk (AURC) per model, without assuming that the semantics of (p) are identical across different models. We acknowledge that we did not explicitly verify perfect cross-model consistency in prompt interpretation or in the exact meaning of the numerical scale; we will make this limitation clearer in the paper and note that, in scenarios where APIs provide logprobs in a comparable way, token-probability–based uncertainty would be a natural extension of our analysis.
>
> 3.  To begin with, it is of course the case—as we already state in §3.2—that the conversations and the base-objective strings are not newly collected, but are drawn from three existing public multi-turn safety datasets. In these datasets, harmful base objectives were originally labeled for other purposes (e.g., jailbreak/safety evaluation), and ObjexMT does not claim to introduce a completely new raw corpus in this sense. Instead, we reinterpret the existing ground-truth base objectives as latent objectives of the multi-turn jailbreak conversations, and build a new task and benchmark configuration around them: single-sentence latent objective extraction plus self-reported confidence, combined with a unified LLM-judge protocol and additional artifacts (model predictions, confidence scores, judge scores). In other words, the existence of ground-truth base objectives is precisely what makes it possible to construct a benchmark that asks “how well can an LLM recognize and extract the harmful objective underlying a dialogue?”, and ObjexMT packages this into a standardized, reproducible evaluation. In the revised version, we will emphasize more clearly in the introduction that ObjexMT is a benchmark built on existing datasets, so as not to give the impression that we collected an entirely new corpus from scratch.
> We also agree that providing more detail on the human calibration set would further strengthen the benchmark, but we do not consider N = 300 expert-labeled items to be “insufficient” for its intended role. These human labels are not used to annotate the entire benchmark; they are used only to calibrate a single similarity→binary threshold for the judge. On these 300 items (≈10.6% of the full 2,817 instances), sweeping the threshold yields (\tau^* = 0.66) with F1 = 0.891, precision = 0.824, and recall = 0.970, indicating a stable decision rule for our downstream analyses. As stated in §3.6, two AI-safety experts produced these consensus labels. We acknowledge that we did not explicitly report inter-annotator agreement prior to consensus; we will add these statistics in the appendix and clearly note that the calibration set can be expanded in future work if needed. Nonetheless, we believe the current setup is adequate for the specific purpose of fixing a single human-aligned threshold.

---

> > ### Author Response · Authors · 2025-11-21
> >
> > 4. We agree that any single-judge design inevitably inherits the semantics of that judge, so in principle the scores could reflect “alignment with GPT-4.1’s internal representation” rather than pure objective-extraction quality. However, in ObjexMT, GPT-4.1 is not used as an unconstrained oracle: we first calibrate its similarity→binary mapping on N = 300 items with human consensus labels (F1 ≈ 0.89, recall ≈ 0.97), and then fix this threshold for all models. Thus, on our data, the reported accuracies are closer to measuring objective-extraction performance under a human-aligned surrogate judge than to arbitrary GPT-4.1 preferences.
> > Moreover, under this human-calibrated decision rule, GPT-4.1’s objective-extraction accuracy is actually lower than that of several other models, so we do not observe a pattern in which the judge systematically favors its own outputs. We will emphasize these empirical observations more clearly in the paper, while still acknowledging that relying on a single LLM judge remains a limitation. We will also highlight that ObjexMT is designed so that future work can re-evaluate the benchmark under alternative or ensemble judges using the released artifacts.
> >
> > 5. We respectfully disagree with the characterization that the paper is “very unfinished.” The current submission already presents a fully specified task and threat model, a fixed evaluation protocol with a human-calibrated judge threshold, statistical tests and effect-size analyses, dataset- and length-based analyses, and explicit limitations and ethics sections. What may feel “unfinished” is not the core methodology but parts of the exposition: in particular, the synthesis in Section 2 and the high-level motivation for focusing on self-reported confidence are intentionally concise. In the revision, we will add short synthesizing and clarifying paragraphs in these places to better connect the five strands in Section 2 and to make our design choices more transparent, while leaving the underlying methods and results unchanged.

---

### Official Review · Reviewer_rFFu · 2025-10-30

**Soundness:** 2
**Presentation:** 2
**Contribution:** 2
**Rating:** 4
**Confidence:** 3

**Summary:**

The paper presents ObjexMT, a benchmark designed to evaluate large language models' (LLMs) ability to extract latent objectives from multi-turn adversarial conversations (e.g., jailbreak prompts) and assess their self-reported confidence (metacognitive calibration). ObjexMT focuses on two key tasks: (i) objective extraction—identifying the core goal from obfuscated multi-turn exchanges, and (ii) calibration—measuring the alignment between self-reported confidence and correctness. Evaluation results reveal that current models demonstrate limited accuracy and significant calibration issues, underscoring the challenges of the LLM-as-a-Judge paradigm.

**Strengths:**

1. ObjexMT tackles a critical challenge in AI safety by formalizing latent objective extraction and confidence calibration.

2. It evaluates six widely used LLMs across diverse datasets, offering valuable insights into model performance and calibration across varying conditions.

**Weaknesses:**

1. Although the paper highlights high-confidence errors, it lacks a detailed taxonomy of failure cases.

2. The writing is difficult to follow, which may hinder understanding.

3. The experiments explore limited methods of self-reporting confidence, making the results less convincing.

4. The evaluation is restricted to six large commercial LLMs, excluding smaller open-source models and safety-tuned variants, which limits the generalizability of the findings.

5. The benchmark's single-sentence objective constraint may oversimplify the complexity of multi-objective or nuanced adversarial prompts.

**Questions:**

1. Could you provide an error analysis to better understand the failure cases?
2. Would you consider presenting results using other methods of self-reporting confidence for a more comprehensive evaluation?

---

### Official Review · Reviewer_Gy5v · 2025-11-01

**Soundness:** 2
**Presentation:** 3
**Contribution:** 3
**Rating:** 4
**Confidence:** 4

**Summary:**

This paper investigates whether LLM-as-a-Judge systems can accurately recover latent objectives from multi-turn jailbreak conversations and reliably calibrate their confidence in such inferences. The researchers propose ObjexMT, a benchmark requiring models to extract a single-sentence base objective from adversarial dialogues and report a confidence score. Using a fixed GPT-4.1 judge for semantic similarity scoring and a human-aligned threshold (τ*=0.66, F1=0.891), they evaluate six state-of-the-art models across 2,817 instances from three datasets.
Results reveal accuracy ranging from 47.4% to 61.2%, with Kimi-K2 achieving the highest extraction accuracy while Claude-Sonnet-4 demonstrates superior calibration (ECE 0.206, Wrong@0.90 14.9%). Critically, accuracy and calibration emerge as independent dimensions. Dataset heterogeneity proves substantial: automated obfuscation in Attack600 yields only 24.3% average accuracy compared to 80.9% for human-authored MHJ dialogues. Transcript length correlates positively with performance, improving from 22-41% accuracy for short dialogues to 74-83% for long ones. Even well-calibrated models exhibit concerning high-confidence errors (14.9-47.7% at confidence ≥0.9), indicating that current LLM judges cannot be safely deployed without human oversight. The findings suggest three recommendations: explicitly surface objectives when possible, implement confidence-based gating for automated decisions, and require human supervision in high-stakes scenarios.재시도Claude는 실수를 할 수 있습니다. 응답을 반드시 다시 확인해 주세요.

**Strengths:**

1. Addresses a critical and timely problem. As LLM-as-a-Judge systems become increasingly deployed in production environments, evaluating their reliability for latent objective inference is essential for AI safety applications.

2. Novel dual-evaluation paradigm. Jointly measuring extraction accuracy and confidence calibration is methodologically innovative and practically valuable. The framework recognizes that opaque judges must signal their own trustworthiness.

3. Comprehensive calibration analysis. Using multiple complementary metrics (ECE, Brier score, Wrong@High-Confidence, AURC) provides a multifaceted assessment of metacognitive reliability rather than relying on a single measure.

4. Rigorous statistical methodology. The paper employs bootstrap confidence intervals (B=10,000), paired significance testing (McNemar), and multiple comparison correction (Holm-Bonferroni), meeting modern standards for statistical rigor.

5. Human-aligned threshold calibration. The use of expert consensus labels (N=300) to determine τ*=0.66 with F1=0.891 provides empirical grounding for the binary classification boundary.

6. Valuable empirical insights on dataset heterogeneity. The finding that dataset construction method drives difficulty more than model choice (24.3% for Attack600 vs 80.9% for MHJ) is substantively important for adversarial robustness research.

7. Orthogonality of accuracy and calibration revealed. Demonstrating that the highest-accuracy model (Kimi-K2, 61.2%) differs from the best-calibrated model (Claude-Sonnet-4, ECE 0.206) is a valuable contribution to understanding LLM capabilities.

8. Excellent transparency and reproducibility. Full release of prompts, data (2,817 instances), human labels, and per-model outputs in structured format enables community validation and extension.

9. Clear practical implications. The Wrong@High-Confidence metric provides immediately actionable deployment thresholds, and recommendations (explicit objective surfacing, confidence gating, human oversight) are concrete.

10. Length-performance relationship documented. The systematic analysis showing monotonic accuracy improvement with transcript length (22-41% for Q1 to 74-83% for Q4) provides operational guidance for risk stratification.

**Weaknesses:**

1. Threshold optimization lacks proper validation. Selecting τ*=0.66 from 101 candidates on the same 300 samples used to report F1=0.891 constitutes a multiple-comparison problem without correction. This risks overfitting; the true generalization performance on held-out data is unknown. The paper needs train-validation splits or cross-validation.

2. Core premise lacks direct empirical validation. The paper claims harmfulness detection differs from intent extraction but only cites prior work without demonstrating this detection-extraction gap on their own data. The paper should show that models correctly identify harmfulness but fail objective extraction on the same instances.

3. Conceptual overreach: "metacognition" overstated. Self-reported confidence scores represent only one narrow aspect of metacognition (confidence in retrieved answer). True metacognition includes explanation of uncertainty sources, recognition of knowledge boundaries, and strategic control. The paper should use the more accurate term "confidence calibration."

4. "LLM-as-a-Judge qualification" overclaimed. Evaluating one specific task (adversarial objective extraction) cannot support broad claims about general judging capability across diverse domains (code review, translation, etc.).

5. Artificial constraints reduce ecological validity. The single-sentence requirement excludes multi-objective attacks and hierarchical intent
structures that may be common in real adversarial scenarios. Temperature T=0 decoding is non-standard for deployment. These choices limit generalizability.

6. Effect size interpretation limited. While Table 7 reports effect sizes (ARR, RR, Cohen's h), the paper doesn't contextualize whether these are practically significant for safety-critical applications. A 12.2% ARR might be crucial or negligible depending on deployment context.

**Questions:**

I thank the authors for their rigorous work on this important problem and for their commitment to transparency through comprehensive artifact release, which will enable the community to build upon these foundational findings.

How do you validate the threshold without overfitting? Please split the 300 labeled samples into 200 training (to optimize τ*) and 100 validation (to report F1), or use 5-fold cross-validation. Current methodology of selecting τ* from 101 candidates on the same data used to report F1=0.891 constitutes a multiple-comparison problem that risks overfitting.

Can you demonstrate the detection-extraction gap on your data? Please evaluate the same models on identical instances for (a) binary harmfulness classification and (b) objective extraction. Report the percentage of cases where models correctly identify harmfulness but incorrectly extract the objective. This would directly validate your core premise that these are distinct capabilities.

What is the judge-dependent variance in model rankings? Please evaluate a subset (N=300-500) using 2-3 alternative judges from different model families (e.g., Claude-3.5-Sonnet, Gemini-2.0-Pro) and report rank correlation (Kendall's τ or Spearman ρ) with GPT-4.1-based rankings. If correlation < 0.8, judge choice substantially affects conclusions and undermines generalizability.

What is the JSON parsing failure rate per model? Please add a column to Table 3 showing the number of valid samples and failure percentage for each model. Additionally, test whether parsing failures correlate with dialogue difficulty metrics (length, turn count, dataset). If models systematically fail on harder cases, this biases accuracy estimates upward.
Critical Transparency Questions
What is the inter-annotator agreement before consensus? For the 300 labeled samples, please report Cohen's κ or Fleiss' κ between the two AI safety experts before they reached consensus. This establishes the reliability ceiling for automated evaluation and indicates whether the task has objective ground truth.

How exactly were the 300 calibration samples selected? Please provide the specific algorithm for "adaptive importance sampling." Was it stratified by dataset, difficulty proxy, or model error patterns? What was the actual distribution (Attack600: 64, 1K: 167, MHJ: 69) compared to the proportions in the full 2,817-instance dataset?

What is the confidence clipping frequency? Models output confidence scores outside [0,1] that require clipping. Please report per-model percentages of predictions requiring clipping and whether this materially affects ECE calculations. High clipping rates suggest models don't follow instructions properly.
Critical Statistical Rigor Questions
Can you decompose Brier scores into components? Please report the Murphy decomposition (Reliability + Resolution - Uncertainty) for each model's Brier score. This distinguishes whether low Brier reflects good calibration or simply high resolution on easy instances, providing deeper insight into calibration quality.

Are results robust to prompt variations? Please test 2-3 alternative extraction prompts (e.g., "identify the core objective" vs. "extract the base prompt") on a subset of 200 instances and report rank correlation. If Spearman ρ > 0.9, robustness is acceptable; if substantially lower, the current results may be prompt-specific rather than measuring true capability.

What are the confidence intervals for Wrong@High-Confidence? Please add Wilson score intervals or bootstrap CIs to the error rates in Table 6 (Wrong@0.80/0.90/0.95). For example, if Claude-Sonnet-4's Wrong@0.90 = 14.9% with CI [11.8%, 18.4%], we can assess whether differences from other models are statistically significant rather than relying on point estimates alone.

---

### Official Review · Reviewer_6LpS · 2025-11-02

**Soundness:** 3
**Presentation:** 3
**Contribution:** 3
**Rating:** 4
**Confidence:** 3

**Summary:**

This paper introduces a new benchmark, ObjexMT.

The benchmark targets multi-turn jailbreak transcripts, where a model must extract a single sentence describing the conversation's objective along with a confidence score.

An LLM judge then computes semantic matching and a confidence score. The semantic matching (i.e., similarity) is based on 300 human annotations. The paper considers four confidence scores: expected calibration error, Brier score, Wrong@[0.80 | 0.90 | 0.95] confidence, and area under the risk–coverage curve.

The paper presents experiments with six models (gpt-4.1, claude-sonnet-4, Qwen3-235B-A22B-FP8, kimi-k2, deepseek-v3.1, gemini-2.5-flash) and three datasets with varying obfuscation levels.

The paper finds that one should be skeptical of the reliability of LLM judges in adversarial contexts.

**Strengths:**

- The paper examines the important topic of using LLMs as judges in safety evaluations.

- ObjexMT addresses a gap in existing jailbreaking benchmarks by extracting a single sentence describing the conversation's objective along with a confidence score.

- Some measures used for the confidence score are intuitive, such as Wrong@0.90 confidence.

- The data sets have different levels of obfuscation.

**Weaknesses:**

- The semantic similarity portion was not fully discussed. Do the experts label semantic similarity and gold objectives independently?

- The wide range of accuracy values was not discussed in detail. What are possible explanations? Would ablation studies help clarify this issue?

- Since the paper does not present any theoretical results, I am not sure how to interpret "claude-sonnet-4 yields the best selective risk and calibration," except that in these experiments it was the "winner." What makes claude-sonnet-4 "better" than the other models? For example, is it its large context window?

**Questions:**

See questions in the Weaknesses text field.

---

### Meta-Review · Area_Chair_JaXc · 2026-01-07

**Summary:**

This paper proposes ObjexMT, a benchmark for evaluating whether LLM-as-a-Judge systems can extract latent objectives from multi-turn jailbreak dialogues and appropriately calibrate their self-reported confidence. Reviewers agree that the problem is timely and relevant for safety evaluation, and that the benchmark surfaces meaningful empirical observations, including frequent high-confidence errors and substantial dataset-dependent difficulty.

However, significant concerns remain regarding the conceptual framing of metacognition, the reliance on a single LLM judge with limited human calibration, and whether the benchmark design and validation are sufficiently rigorous to support the paper's broader claims about LLM-as-a-Judge qualification. Given the high selectivity of the venue, these unresolved issues substantially weaken the case for acceptance.

**Reviewer Concerns:**

Overall, the rebuttal does not meaningfully address the core concerns raised by the reviewers.

**Concerns still outstanding:**
* Threshold selection on the same 300 samples used for reporting performance raises unresolved overfitting and multiple-comparison concerns.
* The framing of self-reported confidence as “metacognition” remains insufficiently justified and is closer to confidence calibration than broader metacognitive ability.
* Lack of reported inter-annotator agreement, limited failure analysis, and dependence on a single judge limit confidence in the benchmark’s robustness.
* Claims about LLM-as-a-Judge qualification extend beyond what is supported by a single task setting and constrained objective format.

**Reviewer Scores:**

Reviewer 6LpS: 4 -> 4
* Reviewers' concerns remain unaddressed by the authors ("no response")

Reviewer Gy5v: 4 -> 4
* Reviewers' concerns remain unaddressed by the authors ("no response")

Reviewer rFFu: 4 -> 4
* Reviewers' concerns remain unaddressed by the authors ("no response")

Reviewer Lrie : 0 -> 0
* The rebuttal does not substantially change this reviewer's view on fundamental methodological and conceptual weaknesses.

**Overall:**
While the paper targets an important and underexplored problem and provides useful empirical observations, the rebuttal does not sufficiently strengthen the methodological foundations or appropriately narrow the conceptual claims. As a result, the work does not meet the acceptance bar for this venue.

---

### Decision · Program_Chairs · 2026-01-26

Reject